# Retrieval of Solar-induced Chlorophyll Fluorescence from Satellite Measurements: Comparison of SIF between TanSat and OCO-2

Lu Yao[1], Yi Liu[1], Dongxu Yang[1], Zhaonan Cai[1], Jing Wang[1], Chao Lin[2], Naimeng Lu[3], Daren Lyu[1], Longfei Tian[4], Maohua Wang[5], Zengshan Yin[4], Yuquan Zheng[2], Sisi Wang[6]

[1]Carbon Neutral Research Center & Key Laboratory of Middle Atmosphere and Global Environment Observation, Institute of Atmospheric Physics, Chinese Academy of Sciences, No. 40, Huayan Li, Chaoyang District, Beijing 100029, China
[2]Changchun Institute of Optics, Fine Mechanics and Physics, Changchun 130033, China
[3]National Satellite Meteorological Center, China Meteorological Administration, Beijing 100081, China
[4]Shanghai Engineering Center for Microsatellites, Shanghai 201203, China
[5]Shanghai Advanced Research Institute, Chinese Academy of Sciences, Shanghai 201210, China
[6]National Remote Sensing Center of China, Beijing 100036, China

*Correspondence to*: Dongxu Yang (yangdx@mail.iap.ac.cn)

**Abstract.** Solar-induced chlorophyll fluorescence (SIF) is emitted during photosynthesis in plant leaves. It constitutes a small additional offset to reflected radiance and can be observed by sensitive instruments with high signal-to-noise ratio and spectral resolution. The Chinese global carbon dioxide monitoring satellite (TanSat) acquires measurements of greenhouse gas column densities. The advanced technical characteristics of the grating spectrometer (ACGS) onboard TanSat enable SIF retrievals from observations in the $O_2$-A band. In this study, one year of SIF data was retrieved from Orbiting Carbon Observatory-2 (OCO-2) and TanSat measurements using the IAPCAS/SIF algorithm. A comparison between the IAPCAS/SIF results retrieved from OCO-2 spectra and the official OCO-2 SIF product (OCO2_Level 2_Lite_SIF.8r) shows a strong linear relationship ($R^2 > 0.85$) and suggests good reliability of the IAPCAS/SIF retrieval algorithm. Comparing global distributions of SIF retrieved by IAPCAS/SIF from TanSat and OCO-2 shows the same spatial pattern for all seasons with gridded SIF difference less than 0.3 W $m^{-2}$ $\mu m^{-1}$ $sr^{-1}$. The global distributions also agree well with the official OCO-2 SIF product with difference less than 0.2 W $m^{-2}$ $\mu m^{-1}$ $sr^{-1}$. The retrieval uncertainty of seasonally gridded TanSat IAPCAS/SIF is less than 0.03 W $m^{-2}$ $\mu m^{-1}$ $sr^{-1}$ whereas the uncertainty of each sounding ranges from 0.1 to 0.6 W $m^{-2}$ $\mu m^{-1}$ $sr^{-1}$. The relationship between annually-averaged SIF products and FLUXCOM gross primary productivity (GPP) was also estimated for six vegetation types in a $1° \times 1°$ grid over the globe, indicating that the SIF data from the two satellites have the same potential in quantitatively characterizing ecosystem productivity. The spatiotemporal consistency between TanSat and OCO-2 and their comparable data quality enable joint usage of the two mission products. Data supplemented by TanSat observations are expected to contribute to the development of global SIF maps with more spatiotemporal detail, which will advance global research on vegetation photosynthesis.

## 1 Introduction

Terrestrial vegetation ecosystems play a large role in the global carbon cycle through the processes of photosynthesis and respiration. Incoming radiation is absorbed, reflected, and/or transmitted by plant leaves. A portion of the absorbed radiation is used by the chlorophyll in plant leaves for carbon fixation, while the rest is either dissipated as heat or re-emitted as solar-induced chlorophyll fluorescence (SIF) at longer wavelengths (Frankenberg et al., 2011a, 2014). In contrast to the traditional remotely sensed vegetation indices obtained from some studies (Frankenberg et al., 2011b; Guanter et al., 2014; Li et al., 2018; Sun et al., 2017a; Yang et al., 2015; Zhang et al., 2014), SIF offers the potential to measure photosynthetic activity and gross primary production (GPP), due to the strong correlation between these measures (Frankenberg et al., 2011b; Guanter et al., 2012, 2014). The fluorescence emission adds a low-intensity radiance of less than 10 W m$^{-2}$ μm$^{-1}$ sr$^{-1}$ and fills in the solar absorption features of the reflected spectrum (Frankenberg et al., 2011a). The filling-in effect of the solar lines (Fraunhofer lines) is the basic principle applied to measure SIF from space using the capabilities of hyperspectral observation (Frankenberg et al., 2011b; Guanter et al., 2012).

The first attempt of observing SIF from space was performed using images acquired by the Medium Resolution Imaging Spectrometer (MERIS) onboard the ENVIronmental SATellite (ENVISAT) (Guanter et al., 2007). This led to a new idea for conducting SIF studies on a global scale. The first global SIF map was retrieved from high-resolution spectra of the Greenhouse-gases Observing SATellite (GOSAT) (Joiner et al., 2011; Frankenberg et al., 2011b). After that, SIF retrievals were implemented for a variety of satellite measurements, such as those from the Global Ozone Monitoring Experiment-2 (GOME-2) instruments onboard meteorological operational satellites, SCIAMACHY onboard ENVISAT, and Orbiting Carbon Observatory-2 (OCO-2) (Joiner et al., 2016; Köhler et al., 2015). The TROPOspheric Monitoring Instrument (TROPOMI) onboard Sentinel 5 Precursor (S-5P) provides more efficient SIF observations in terms of global coverage and new opportunities for exploring the application potential of SIF data in the terrestrial biosphere as well as in climate research (Doughty et al., 2019; Köhler et al., 2018b). Furthermore, an upcoming European Space Agency mission called FLuorescence EXplorer (FLEX), the first satellite dedicated to SIF observation, will launch in the middle of 2024 (Drusch et al., 2017). Many studies on SIF applications have been initiated with the accumulation of SIF products in recent years. The responses of satellite-measured SIF to environmental conditions have been applied to drought dynamics monitoring and regional vegetation water stress estimation (Lee et al., 2013; Sun et al., 2015; Yoshida et al., 2015). As a proxy of photosynthesis, SIF acts as a powerful constraint parameter in estimating carbon exchange in an ecosystem between the atmosphere, ocean, and soil; as such, the analysis of the relationship between SIF and GPP has become an important research topic (Li et al., 2018; Köhler et al., 2018a; Sun et al., 2017a; Zhang et al., 2018). The strong linear relationship between them paves the way for improving terrestrial ecosystem model simulations of GPP, along with consequent improvement of global carbon flux estimation (MacBean et al., 2018; Yin et al., 2020). GPP estimations based on satellite-measured SIF have proven to be an effective method validated by in-situ flux observations (Joiner et al., 2018; Qiu et al., 2020). However, uncertainty in the factors that determine the relationship between SIF and GPP still exists and is a key limitation in the application of SIF to flux estimation. Based on

multi-satellite SIF products, eddy covariance flux tower observations, and ecological models, the relationship between SIF and
GPP under different environmental conditions has been discussed in a number of studies to analyze the dominant factors for
the growing status of different biomes, such as temperature, soil moisture, and vegetation types (Chen et al., 2020; Doughty et
al., 2019; Li et al., 2020; Qiu et al., 2020; Yin et al., 2020).
The Chinese global carbon dioxide monitoring satellite (TanSat) was launched in December 2016. Aiming at acquiring $CO_2$
concentrations similar to OCO-2, TanSat flies in a sun-synchronous orbit at approximately 700 km height with a 16-day repeat
cycle and an equator crossing time of ~1:30 p.m. local time (Cai et al., 2014; Liu et al., 2018; Yang et al., 2018). Onboard
TanSat, the hyperspectral Atmospheric Carbon-dioxide Grating Spectrometer (ACGS) is designed to separately record solar
backscatter spectra in three channels centered at 0.76 μm ($O_2$-A band), 1.61 μm (weak $CO_2$ absorption band), and 2.06 μm
(strong $CO_2$ absorption band). Many Optimal Estimation Method (OEM) full physics retrieval algorithms have been developed
and applied for the total column-averaged dry air CO2 mole fraction ($XCO_2$) retrievals (Bösch et al., 2006; O'Dell et al., 2012;
Reuter et al., 2010; Yang et al., 2015b; Yoshida et al., 2011, 2013). The Institute of Atmospheric Physics Carbon Dioxide
Retrieval Algorithm for Satellite Remote Sensing (IAPCAS) algorithm has been applied for TanSat XCO2 retrievals (Yang et
al., 2018; Yang et al., 2021) and was also previously tested on spectra from the GOSAT and OCO-2 missions (Yang et al.,
2015b). However, the fluorescence feature causes substantial biases when retrieving surface pressure and scattering parameters
from the $O_2$-A band, and the associated errors propagate into the $XCO_2$ retrievals. In previous $XCO_2$ retrievals, the surface
emissions were well modeled as a continuum offset of the $O_2$-A band to reduce errors (Frankenberg et al., 2011a, 2012; Joiner
et al., 2012). For TanSat, its high spectral resolution of ~0.044 nm and a signal-to-noise ratio of ~360 in the $O_2$-A band makes
it possible to obtain SIF, with a spatial resolution of 2 km × 2 km in nadir mode (Liu et al., 2018).
Various approaches have been used to infer SIF from satellite measurements (Frankenberg et al., 2011b, 2014a, 2014b; Guanter
et al., 2007, 2012, 2015; Joiner et al., 2011, 2013, 2016; Köhler et al., 2015, 2018b). The SIF signal induces a filling-in effect
of solar lines, which can be used for SIF retrieval, as the fractional depth of solar Fraunhofer lines does not change during
radiation transmission in the atmosphere. To be able to measure the filling-in features from SIF, high-resolution spectra are
required to describe subtle changes in the spectral absorption lines. Given highly resolved spectral features, a method was
developed based on solar line fitting and the Beer-Lambertian law. This method is robust and accurate when the spectrum is
out of the influence of telluric absorptions, even in the presence of aerosols (Frankenberg et al., 2011a; Joiner et al., 2011); in
the current study, this method was applied to develop the IAPCAS/SIF algorithm. Another SIF retrieval method is the data-
driven algorithm based on the singular value decomposition (SVD) technique (Joiner et al., 2011; Guanter et al., 2012), which
has been broadly applied to GOSAT, OCO-2, TanSat and TROPOMI (Joiner et al., 2011; Guanter et al., 2012, 2015;
Frankenberg et al., 2014a; Du et al., 2018; Köhler et al., 2018b). In the data-driven method, the spectrum is represented as a
linear combination of the SIF signal and several singular vectors that are trained from non-fluorescent scenes by SVD; thus,
the SIF signal can be obtained with linear least-squares fitting  (Du et al., 2018; Guanter et al., 2012). The first TanSat SIF
map was obtained by the SVD method (Du et al., 2018). In a previous study, a new TanSat SIF product retrieved by
IAPCAS/SIF algorithm was introduced and the two kinds of TanSat SIF products by IAPCAS/SIF and the SVD methods were
compared (Yao et al., 2021). The preliminary comparison between the two TanSat SIF products showed that the two SIF
products share a similar global pattern and signal magnitude for all seasons while different biases still exist in four seasons
(Yao et al., 2021). The different biases in the four seasons may be caused by the different training samples of the SVD method,
which indicates that the training samples have a significant impact on the retrieval results. In order to obtain stable SIF data
products from TanSat and other subsequent satellite missions, it is particularly important to establish a stable and high-
precision SIF inversion algorithm. To further validate the IAPCAS/SIF algorithm and to test the potential for synergistic,
multi-satellite SIF analysis, in this study, we detail the IAPCAS/SIF algorithm for TanSat and we compare the SIF products
from TanSat and OCO-2 for a range of spatiotemporal scales.

## 110 2 Data and retrieval algorithm

### 111 2.1 Retrieval Principle and Method

We used TanSat version 2 Level 1B (L1B) nadir-mode earth observation data in the retrieval process. The measurements
covered the period from March 2017 to February 2018. Polarized radiance in the $O_2$-A band with a spectral resolution of 0.044
nm was provided in the L1B data, and two micro-windows near 757 nm (758.3-759.2 nm) and 771 nm (769.6-770.3 nm) were
chosen to retrieve top-of-atmosphere (TOA) SIF while avoiding the contamination from strong lines of atmospheric gas
absorption. The retrieval was independent for each micro-window as shown in Figure 1. To avoid duplication of information,
we use the SIF product at 757 nm as the example in the analysis.

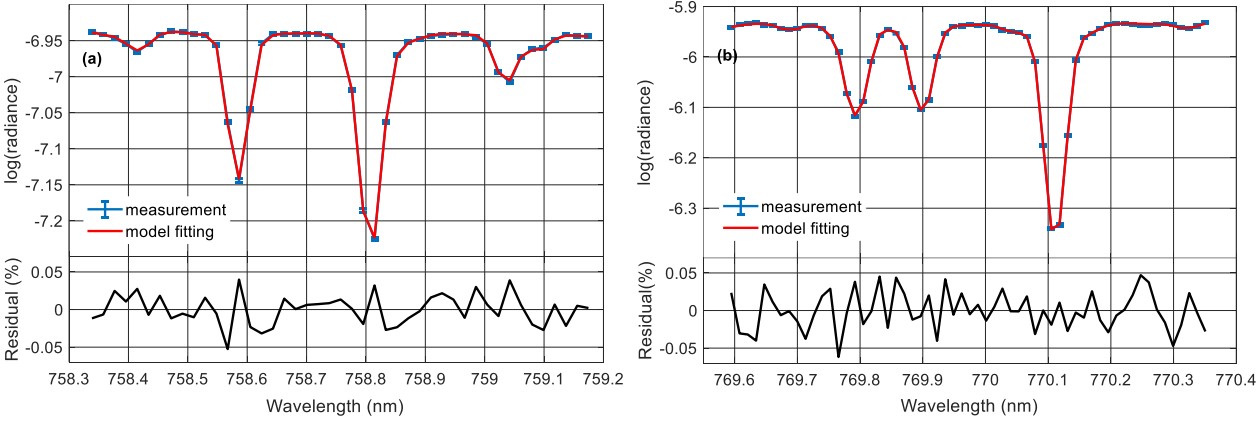

**Figure 1: The fitted spectra and residuals for the (a) 757 nm and (b) 771 nm micro-windows of TanSat measurement. The error bar**
**of the measured spectra depicts the estimated precision of each TanSat sounding.**
Filling-in on solar lines by chlorophyll fluorescence in the $O_2$-A band can be detected in the hyperspectral measurements from
TanSat. This effect on spectral radiance is different from the impact of atmospheric and surface processes, e.g., scattering and
absorption. For example, scattering by aerosols and clouds does not change the relative depth of clear solar lines, unlike the
SIF emission signal. We applied the differential optical absorption spectroscopy (DOAS) technique to IAPCAS/SIF algorithm
for TanSat measurement (Frankenberg, 2014b; Sun et al., 2018).
The TOA spectral radiance ($L_{TOA}^\lambda$) at wavelength $\lambda$ can be represented as follows:
$$L_{TOA}^\lambda = I_t^\lambda \cdot \mu_0 \cdot (\rho_0^\lambda + \frac{\rho_s^\lambda \cdot T_\downarrow^\lambda \cdot T_\uparrow^\lambda}{\pi}) + F_{TOA}^\lambda \tag{1}$$

where $I_t^\lambda$ is the incident solar irradiance at the TOA, $\mu_0$ is the cosine of the solar zenith angle (SZA), $\rho_0^\lambda$ is atmospheric path
reflectance, $\rho_s^\lambda$ is surface reflectance, and $T_\downarrow^\lambda$ and $T_\uparrow^\lambda$ are the total atmospheric transmittances along the light-path in the
downstream and upstream directions, respectively. $F_{TOA}^\lambda$ is the SIF radiance at TOA.
The first term on the right of Eq. (1) represents the transmission process of solar radiance. In the micro-windows used in SIF
retrieval, gas absorption is very weak and smooth, and hence, the atmosphere term $\mu_0 \cdot (\rho_0^\lambda + \frac{\rho_s^\lambda \cdot T_\downarrow^\lambda \cdot T_\uparrow^\lambda}{\pi})$ can be simplified to a
low-order polynomial $\sum_{i=0}^n a_i \cdot \lambda^i$ that varies with $\lambda$ (Joiner et al., 2013; Sun et al., 2018); this is always valid as long as the
spectrum fitting range is out of sharp atmospheric absorptions. Then Eq. (1) could be represented as:
$$L_{TOA}^\lambda(F_{TOA}^\lambda, \boldsymbol{a}) = < I_t^\lambda > \cdot \sum_{i=0}^n a_i \cdot \lambda^i + F_{TOA}^\lambda \tag{2}$$

where $< >$ denote the convolution with the ISRF from line-by-line spectra, and the coefficient vector $\boldsymbol{a}$ determines the
wavelength dependence polynomial for the atmosphere term.
To facilitate the extraction of SIF signals, the radiance is normalized to the continuum level radiance and the relative
contribution of SIF to the continuum level radiance $F_s^{rel}$ is defined. In the micro-window, SIF was regarded as a constant signal
due to its small changes. When the spectral radiance measurement was converted to logarithmic space, the forward model was
expressed as:
$$f(F_s^{rel}, \boldsymbol{b}) = log(< I_t + F_s^{rel} >) + \sum_{i=0}^n b_i \cdot \lambda^i \tag{3}$$

Where $I_t$ is a normalized disk-integrated solar transmission model. The vector $\boldsymbol{b}$ consists of the polynomial coefficients $b_i$
and we used a second-order polynomial ($i = 0, 1, 2$) in the retrieval.
Although the atmospheric gas absorption was very weak in the micro-window, the weak absorption and the far-wing effects
($O_2$ lines) can still change spectral features, which induces errors in spectrum fitting. In other physics-based retrievals, the
surface pressure data of the European Centre for Medium-Range Weather Forecasts (ECMWF) together with topographic data
are usually used as the true surface pressure to simulate the atmospheric transmission in the range of the $O_2$-A band. However,
there is still a difference between the true surface pressure and the model surface pressure, so we introduced a factor here to
reduce the influence of the inaccurate surface pressure. In the IAPCAS/SIF algorithm, we use the ECMWF interim surface
pressure ($0.75° \times 0.75°$) to estimate $O_2$ absorption first and then modify the absorption feature by a scale factor. The scale
factor is obtained simultaneously in SIF retrieval to reduce the error induced by the uncertainty in surface pressure. As
described by Yang (2020), there is also a continuum feature in TanSat L1B data that needs to be considered for the high-quality
fitting of the $O_2$-A band. However, in this study, this continuum feature was not corrected, as the impact of such a smooth
continuum variation in the micro-window is weak and the polynomial continuum model is capable of compensating for most
of this effect.
The state vector includes the relative SIF signal $F_s^{rel}$, a wavenumber shift, the scale factor for the $O_2$ column absorption, and
coefficients of the polynomial. The continuum level radiance $I_{cont}$ within the fitting window is calculated using the radiance
outside the absorption features in the micro-window and is then used for the actual SIF signal calculation thus: $F = F_s^{rel} \cdot I_{cont}$.
In the IAPCAS/SIF algorithm, we used an OEM for state vector optimization in the retrieval process. Compared to the IAPCAS
$XCO_2$ retrieval, the IAPCAS/SIF retrieval employs a state vector with fewer elements and a much simpler forward model, so
there is no need to perform complex radiative transfer calculations. Considering the low complexity of SIF retrieval, the Gauss-
Newton method was applied iteratively to find the optimal solution.
**2.2 Bias Corrections**
A systematic error remains in the raw SIF retrieval output if no bias correction is performed; similar results have been reported
in GOSAT and OCO-2 SIF retrieval studies (Frankenberg et al., 2011a, 2011b; Sun et al., 2018). This is because the SIF signal
is weak (e.g., typically ~1-2% of the continuum level radiance), which means that even a small issue in the measurement, such
as a zero-offset caused by radiometric calibration error, could induce significant bias. Unfortunately, the lack of knowledge on
in-flight instrument performance makes it difficult to perform a direct systematic bias correction in the measured spectrum.
The bias was considered to be related to the continuum level radiance in the previous works. To get the relationship between
the continuum level radiance and the bias, we calculated the mean bias for continuum level radiance at the interval of 5 W m$^{-2}$
µm$^{-1}$ sr$^{-1}$ from all non-fluorescence measurements, and then a piecewise linear function fit was applied to describe the
relationship between the continuum level radiance and the biases.
The non-fluorescence soundings that were used in the bias estimation were based on the dataset "sounding_landCover" in
TanSat L1B data. This dataset depends on the MODIS land cover product and provides a scheme consisting of 17 land cover
classifications defined by the International Geosphere-Biosphere Programme. The measurements marked as "snow and ice,"
"barren," and "sparsely vegetated" were chosen to estimate the bias. Calibrations compensated for most of the instrument
degradations, but this alone was not perfect. To reduce the impact from the remaining minor discrepancies, we built the bias
correction function daily to obtain bias for each sounding via interpolation of the continuum level radiance (Sun et al., 2017b,
181 2018).

The bias curves shown in Figure 2 differ significantly between TanSat and OCO-2. This is mostly due to the differences in
instrument performance and radiometric calibration. In general, the TanSat bias curves exhibited two peaks at radiance levels
of approximately 40 and 125 W m$^{-2}$ µm$^{-1}$ sr$^{-1}$, separately, and most biases were larger than 0.015. For OCO-2, the curves
dropped sharply at low radiance levels, reaching the valley at a radiance level of approximately 40 W m$^{-2}$ µm$^{-1}$ sr$^{-1}$, and then
increased slowly with the radiance level.

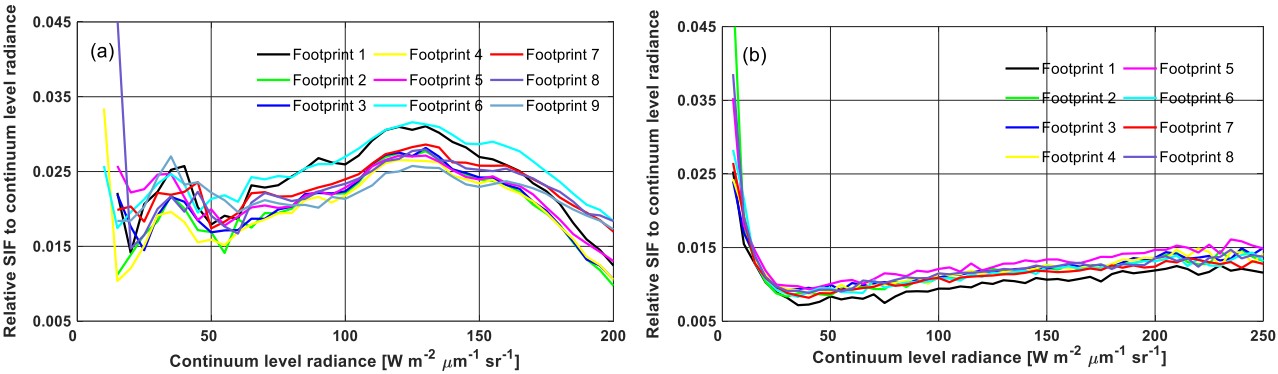


**Figure 2: Variations in the bias correction curves of continuum level radiance from (a) TanSat on July 7, 2017, and (b) Orbiting**
**Carbon Observatory-2 (OCO-2) on June 16, 2017. The different colors in the legend present different footprints of the satellite frame.**
**2.3 Data Quality Controls**
Only data that passed quality control were used in further applications. There were two data quality control processes for the
SIF products: pre-screening and post-screening. Pre-screening focused mainly on cloud screening; only cloud-free
measurements were used in SIF retrieval. A surface pressure difference (SPD), defined as:
$\Delta P_0 = |P_{retrieval} - P_{ECMWF}|$ (3)
was used to evaluate cloud contamination along with a chi-square test
$\chi^2 = \sum \frac{(y_{sim} - y_{obs})^2}{y_{noise}^2}$ (4)
where $y_{sim}$, $y_{obs}$, and $y_{noise}$ represent the model fitting spectrum, observation spectrum, and spectrum noise, respectively.
$P_{retrieval}$ is the apparent surface pressure obtained from $O_2$-A band surface pressure retrieval, assuming a Rayleigh-scattering
atmosphere. $P_{ECMWF}$ is the surface pressure data from the ECMWF interim (0.75° × 0.75°) reanalysis data product (Dee et al.,
2011), which is interpolated to the sounding location and corrected for elevation differences with the Shuttle Radar Topography
Mission Global 30 Arc-Second Elevation digital elevation model (DOI: 10.5067/MEaSUREs/SRTM/SRTMGL30.002). A
"cloud-free" measurement was required to simultaneously satisfy an SPD of less than 20 hPa and a $\chi^2$ value of less than 80.
Here, post-screening was applied to filter out "bad" retrievals; this screening process involved the following steps: (1) SIF
retrievals with reduced $\chi^2$ ($\chi^2_{red}$) values ranging from 0.7 to 1.3 were considered "good" fitting, (2) continuum level radiance
outside the range of 15 ~ 200 W m$^{-2}$ μm$^{-1}$ sr$^{-1}$ was screened out to avoid scenes too bright or too dark, and (3) soundings with
the SZA higher than 60° were also filtered out.

**2.4 IAPCAS versus IMAP-DOAS OCO-2 SIF Retrieval**
Before applied to TanSat retrievals, we tested the IAPCAS/SIF algorithm on the OCO-2 L1B data first
(OCO2_L1B_Science.8r) and then compared the retrieval results with the OCO-2 L2 Lite SIF product (OCO2_Level
2_Lite_SIF.8r) retrieved by the Iterative Maximum A Posteriori-Differential Optical Absorption Spectroscopy (IMAP-DOAS)
algorithm (Frankenberg, 2014b). The Lite product provides the SIF value for each sounding and hence the SIF comparison
could be performed on the sounding scale for each month.
Table 1 displays the relationship of OCO-2 SIF values between the IAPCAS/SIF and IMAP-DOAS at 757 nm micro-window
for each month. Overall, the two SIF products were in good agreement. The linear fitting of the two SIF products suggests that
they are highly correlated, as indicated by the strong linear relationship with $R^2$ mostly larger than 0.85 and the root mean
square error (RMSE) of about 0.2 W m$^{-2}$ μm$^{-1}$ sr$^{-1}$. Good consistency between the two SIF products implies the reliability of
the IAPCAS/SIF algorithm; thus, it was further applied to TanSat SIF retrieval. However, there was still a small bias in the
comparisons, which was due, most likely, to the impact of differences in the bias correction method, retrieval algorithm, and
fitting window.
**Table 1:** Summary of the relationship between the IAPCAS OCO-2 and IMAP-DOAS OCO-2 solar-induced chlorophyll fluorescence
(SIF) products at 757nm micro-window.

| month | Number of soundings | Slope | Intercept | $R^2$ | RMSE/ W m$^{-2}$ μm$^{-1}$ sr$^{-1}$ |
|---|---|---|---|---|---|
| 2017/03 | 1097277 | 0.85 | 0.034 | 0.86 | 0.18 |
| 2017/04 | 1119464 | 0.86 | 0.045 | 0.87 | 0.19 |
| 2017/05 | 1054235 | 0.88 | 0.041 | 0.88 | 0.19 |
| 2017/06 | 1014848 | 0.91 | 0.032 | 0.90 | 0.19 |
| 2017/07 | 965309 | 0.92 | 0.011 | 0.91 | 0.19 |
| 2017/09 | 211219 | 0.88 | 0.005 | 0.81 | 0.23 |
| 2017/10 | 473359 | 0.88 | 0.031 | 0.88 | 0.17 |
| 2017/11 | 579009 | 0.87 | 0.022 | 0.85 | 0.19 |
| 2017/12 | 645134 | 0.87 | 0.020 | 0.88 | 0.16 |
| 2018/01 | 788655 | 0.87 | 0.019 | 0.88 | 0.17 |
| 2018/02* | 629995 | 0.86 | 0.024 | 0.87 | 0.18 |

* Due to the lack of OCO-2 measurements in August 2017, the comparison is only performed for 11 months.

## 3 Results and Discussion

### 3.1 Comparison between TanSat and OCO-2 SIF Measurements

The comparison between TanSat and OCO-2 SIF measurements is a useful and powerful method for further verification of the
IAPCAS/SIF algorithm. The reason for adopting OCO-2 data is that OCO-2 and TanSat have similar observation modes,
including scanning method, transit time, spatial resolution, spectral resolution, and spectral range. The similarities mean that
the SIF product from the two satellite missions can be directly compared. Directly comparing OCO-2 and TanSat SIF
measurements could provide information on joint data application at the sounding scale for further studies. However, an
identical sounding overlap barely exists because the two satellites often have different nadir tracks on the ground, which is
induced by the different temporal and spatial intervals of the two satellite missions. Fortunately, the ground tracks of the two
satellites were relatively close from April 17 to April 23, 2017. A couple of overlapping orbits were found in the measurements
obtained from Africa with the orbit number of 1733 from TanSat and 14890a from OCO-2 (Figure 3). In the comparison, the
OCO2_Level 2_Lite_SIF.8r product was used to present the SIF emission over the study area. These overlapping
measurements encompassed multiple land cover types, in which the SIF varied within an acceptable time difference (<5 min).
Overall, measurements from the two satellites indicated SIF variation with land cover type. The SIF emission over evergreen
broadleaf forests was larger than that over savannas, and grasslands exhibited the lowest SIF emission in April (Figure 3a,b).
The mean SIF emission over evergreen broadleaf forests was approximately 0.9-1.1 W $m^{-2}$ $\mu m^{-1}$ $sr^{-1}$, whereas those over
savannas and grasslands were 0.5-0.7 W $m^{-2}$ $\mu m^{-1}$ $sr^{-1}$ and less than 0.1 W $m^{-2}$ $\mu m^{-1}$ $sr^{-1}$, respectively (Figure 3c,d).
Furthermore, we also found a significant difference in the SIF emission intensity over tropical savannas, which was observed
by both satellites (Figure 3c,d).


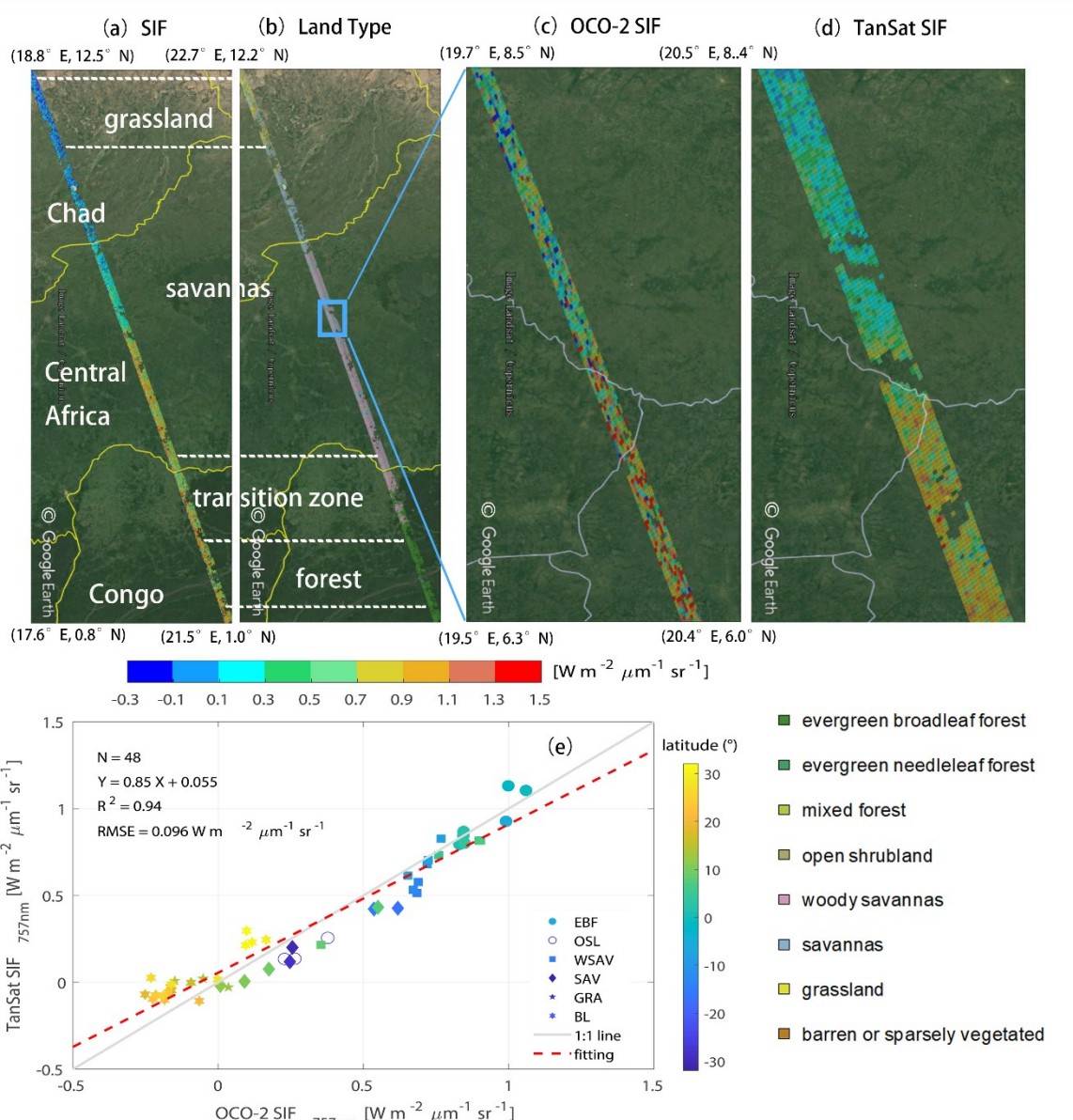

**Figure 3: Overlapping orbits of TanSat and OCO-2 on April 19, 2017 over Africa displayed in Google Earth, (a) the SIF measurements from both satellites and (b) the footprint land cover type were compared. Compared to OCO-2, TanSat has a wider swath width. A zoom-in view over savannas shows variations in the SIF signal measured by (c) OCO-2 and (d) TanSat. The land surface image shown in Google earth is provided by Landsat/Copernicus team. Following the International Geosphere-Biosphere Programme classification scheme, the vertical legend on the bottom right corner depicts the land cover type that occurs in the study area. The middle horizontal color bar represents the intensity of the SIF radiance. (e) Small-area SIF comparison between OCO-2 and TanSat; each data point represents the mean SIF of a degree in latitude (colors) along the track. The marker legend that is shown on the bottom right of the plot indicates the dominant land cover (defined as the majority land cover type of each sounding) in each small area. There are six land cover types including evergreen broadleaf forest (EBF), open shrubland (OSL), woody savanna (WSAV), savanna (SAV), grassland (GRA), and barren land (BL). The red dashed line represents the linear fit between the two SIF products with statistics shown in the upper left of the plot. The gray line indicates a 1:1 relationship for reference.**


Because the footprint sizes of the two satellites are different, it is difficult to make a direct footprint-to-footprint comparison.
Therefore, we made the comparison between the two satellite measurements based on a small area average. Each small area
spans a degree in latitude and continues along the track. The small area-averaged SIF comparison is shown in Figure 3e. The
results indicate good agreement, with an $R^2$ of 0.94 and an RMSE of 0.096 W m$^{-2}$ μm$^{-1}$ sr$^{-1}$. Additional ground-based SIF
measurement setups (Guanter et al., 2007; Liu et al., 2019; van der Tol et al., 2016; Yang et al., 2015a; Yu et al., 2019) should
allow for direct evaluation of satellite retrieval accuracy in the future.




**Figure 4: Global TanSat SIF (left, a-d), differences between TanSat and IAPCAS OCO-2 SIF values (middle, e-h), and the grid-cell**
**retrieval uncertainty estimated from TanSat (right, i-l) at 1° × 1° spatial resolution. The maps in each row represent a Northern**
**Hemisphere season, i.e., spring (MAM), summer (JJA), fall (SON), and winter (DJF).**

Figure 4 shows the global SIF comparison between IAPCAS/SIF retrieved from OCO-2 and TanSat; this comparison is only
performed at $1° \times 1°$ spatial resolution. In general, the difference in SIF globally is mostly less than 0.3 W m$^{-2}$ μm$^{-1}$ sr$^{-1}$ for
all seasons, and on average, the smallest difference appears in fall. There are regional biases observed in North Africa, South
Africa, South America, and Europe in all seasons except fall. This is mainly caused by the differences in instrument
performance between TanSat and OCO-2, such as the Instrument Spectral Respond and the Signal-to-Noise. The instrument
performance difference is represented by the different structural characteristics of the bias curves. The bias correction
compensates for most of the bias caused by instrument performance; however, small biases could remain. Furthermore, the
hundreds of kilometers of distance between the OCO-2 and TanSat footprints, for example, over different vegetation regions,
will also cause some measurement discrepancies. The global distribution of the two satellites was also compared with the
official OCO-2 SIF data on the global scale, the results show that the difference between the retrieved SIF maps and the official
map is less than 0.2 W m$^{-2}$ μm$^{-1}$ sr$^{-1}$, indicating that the retrieved SIF data from OCO-2 and TanSat both have good SIF
characterization capabilities on a global scale. The uncertainty σ of each sounding was estimated to validate SIF reliability and
is provided in the product. σ is derived from the retrieval error covariance matrix, $S_e = (K^T S_0^{-1} K)^{-1}$, where $K$ is the Jacobian
matrix from the forward model fitting and $S_0$ is the measurement error covariance matrix that is calculated from the instrument
spectrum noise. In general, σ ranges from 0.1 to 0.6 W m$^{-2}$ μm$^{-1}$ sr$^{-1}$ for both TanSat and OCO-2 measurements in the 757
nm fitting window, which is of a similar magnitude and data range as those of previous studies (Du et al., 2018; Frankenberg
et al., 2014a). Meanwhile, the standard error of the mean SIF in each grid $\sigma_{meas}$ was estimated to represent the gridded retrieval
error and natural variability, which is calculated from TanSat SIF values with $\sigma_{meas} = \frac{\sigma_{std}}{\sqrt{n}}$ and $\sigma_{std} = \sqrt{\frac{\sum_{i=1}^{n}(SIF_i - \overline{SIF})^2}{n}}$, where
$\sigma_{std}$ represents the standard deviation of the grid cell with $n$ soundings, $SIF_i$ is the retrieved SIF values of each sounding,
and $\overline{SIF}$ is the mean SIF value for all measurements in the grid. As depicted in the right column of Figure 4, the $\sigma_{meas}$ of each
grid cell is much lower than the precision of a single sounding. The $\sigma_{meas}$ for South America is larger than that for any other
region on the globe (Figure 4i-l). This is similar to that of OCO-2 SIF retrieval and caused by fewer effective measurements
due to the South Atlantic Anomaly (Sun et al., 2018). The difference in SIF emission values between the two satellites indicates
that the synergistic use of two satellite SIF products still requires analysis of the impact of instrument differences, although
the two satellite SIF products share the same spatiotemporal pattern on a global scale.

## 3.2 SIF Global Distribution and Temporal Variation

The SIF emission intensity reflects the growth status of vegetation, and hence the overall global vegetation status can be
represented by global SIF maps for each season. TanSat SIF over a whole year's cycle, from March 2017 to February 2018, is
represented seasonally as a $1° \times 1°$ grid spatially. The seasonal variation in SIF emission is clear in the Northern Hemisphere,
i.e., it is enhanced from spring to summer and then decreases (Sun et al., 2018).
In general, the SIF emission varied with latitude and the vegetation-covered areas near the equator maintained a continuous
SIF emission throughout the year. Large SIF emissions in the Northern Hemisphere, above 1.5 W m$^{-2}$ μm$^{-1}$ sr$^{-1}$, mostly from
the eastern U.S., southeast of China, and southern Asia in summer, were due to the large areas of cropland. There was also an
obvious SIF emission of 1-1.2 W m$^{-2}$ μm$^{-1}$ sr$^{-1}$ observed over Central Europe and northeastern China during the summer. In
these regions, croplands and deciduous forests contribute to SIF emissions. In the Southern Hemisphere, the strongest SIF
emission occurred in the Amazon, with a level of approximately 1-2 W m$^{-2}$ μm$^{-1}$ sr$^{-1}$ in DJF (Northern Hemisphere winter),
where there is an evergreen broadleaf rainforest. Africa, which is covered by evergreen broadleaf rainforests and woody
savannas, had an average SIF value of 0.7-1.5 W m$^{-2}$ μm$^{-1}$ sr$^{-1}$ during the year.
The SIF-GPP relationship over different vegetation types was also investigated by comparing the annual mean satellite SIF
measurements with the FLUXCOM GPP (Jung et al., 2020; Tramontana et al., 2016) dataset in a 1° × 1° grid over the globe.
The FLUXCOM GPP dataset used in the study comprises monthly global gridded flux products with remote sensing and
meteorological/climate forcing (RS+METEO) setups, which are derived from mean seasonal cycles according to MODIS data
and daily meteorological information (Jung et al., 2020; Tramontana et al., 2016). In the correlation analysis, the high spatial
resolution (0.5°× 0.5°) of the FLUXCOM GPP was first resampled to 1°× 1° to keep the same temporal-spatial scale of SIF
and GPP data. The satellite-measured SIF is an instantaneous emission signal that varies with incident solar radiance within
the day. To reduce the differences caused by the observation time and SZA at different latitudes, we applied a daily adjustment
factor to convert the instantaneous SIF emission into a daily mean SIF (Du et al., 2018; Frankenberg et al., 2011b; Sun et al.,
2018). The daily adjustment factor d is calculated as follows:
$$d = \frac{\int_{t=t_0-12h}^{t=t_0+12h} cos\,(SZA(t)) \cdot dt}{cos(SZA(t_0))} \tag{5}$$
where $t_0$ is the observation time in fractional days and $SZA(t)$ is a function of latitude, longitude, and time for calculating the
SZA of the measurements. The annual averaged SIF is calculated from the daily mean SIF. To evaluate the relationship
between SIF and GPP on the periodic scale of vegetation growth status, annually-averaged data were used in the regression
fitting analysis.

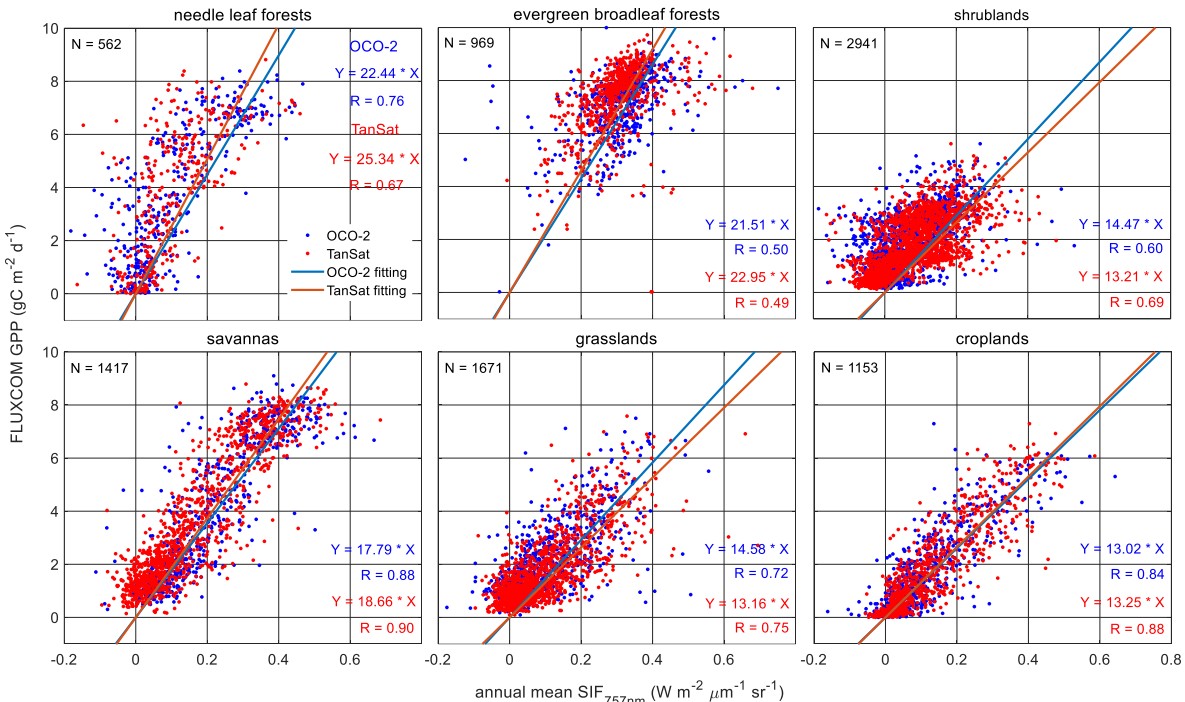


**Figure 5: Relationship between annual mean SIF and FLUXCOM gross primary production (GPP) from March 2017 to February 2018. Blue and red dots represent OCO-2 and TanSat SIF grids, respectively. Fitted lines and statistics for OCO-2 and TanSat are shown in each plot.**

Figure 5 shows the linear fits for six vegetation types, including needle leaf forest, evergreen broadleaf forest, shrubland, savanna, grassland, and cropland. Recent studies have shown a strong linear correlation between SIF and GPP. The TanSat SIF and the OCO-2 official SIF data were used to estimate the SIF-GPP correlation. To make a direct comparison of the relationship between SIF and GPP among various vegetation types, we used non-offset linear fitting to indicate the correlation between satellite SIF and FLUXCOM GPP. For savanna and cropland, there were strong relationships between the mean SIF and GPP with an R-value above 0.84. The fitting results show that the SIF products of the two satellites have similar capabilities in characterizing GPP, especially for the evergreen broadleaf forest, savanna, and cropland, with slopes of approximately 21, 18, and 13, respectively. For shrubland and grassland, the slope of OCO-2 SIF with GPP is higher than that of TanSat and has a worse correlation. For forests, OCO-2 SIF presents a better correlation with GPP, especially in the needle leaf forest. As a whole, for the same vegetation type, the SIF-GPP correlations for the two satellites are rather similar, indicating that the two satellite SIF products have similar capabilities in characterizing GPP. It shows the strong feasibility of the comprehensive application of different satellite SIF products. For different vegetation types, the SIF-GPP correlations were significantly different, indicating the different ability of SIF to characterize GPP of different vegetation. It represents that vegetation type is a key factor in determining the SIF-GPP relationship. The markedly different fitting slopes across various biomes suggest that the application of SIF in GPP estimation needs more detailed analysis although the evidence of the strong linear relationship between them.

## 4 Conclusions

In this paper, we introduced the retrieval algorithm IAPCAS/SIF and its application in TanSat and OCO-2 measurements. One year (March 2017-February 2018) of TanSat SIF data was introduced and compared with OCO-2 measurements in this study. The TanSat and OCO-2 SIF products based on the IAPCAS/SIF algorithm are available on the Cooperation on the Analysis of carbon SAtellites data (CASA) website, www.chinageoss.org/tansat. Comparing TanSat and OCO-2 measurements directly, using a case study, and indirectly, with global $1°\times1°$ grid data, showed consistency between the two satellite missions, indicating that the coordinated usage of the two data products is possible in future studies. The correlation analysis between SIF and GPP further verified the feasibility of the synergistic application of SIF products from different satellite missions. Meanwhile, it should be noticed that the difference in the ability of satellite SIF products to characterize different vegetation types in data applications. With more satellites becoming available for SIF observations, space-based SIF observations have recently expanded in range to provide broad spatiotemporal coverage. The next-generation Chinese carbon monitoring satellite (TanSat-2) is now in the preliminary design phase, which is designed to be a constellation of six satellites to measure different kinds of greenhouse gases and trace gases in a more efficient way, including $CO_2$,$CH_4$, CO, NOx, as well as SIF. SIF measurements from TanSat-2 will provide global data products over broader coverage areas with less noise. The improvement in the spatiotemporal resolution of SIF data will benefit GPP predictions based on the numerous studies of the linear relationship between SIF and GPP. In future work, the measurement accuracy should be validated directly using ground-based measurements to ensure data quality.

**Data availability**

The SIF products of TanSat and OCO-2 by IAPCAS/SIF algorithm are available on the Cooperation on the Analysis of carbon SAtellites data (CASA) website (http://www.chinageoss.cn/tansat/index.html).

**Author contributions**

L.Y. and D.Y. developed the retrieval algorithm, designed the study, and wrote the paper. Y.L. led the SIF data process and analysis. Y.L., D.Y., Z.C., and J.W. contributed to manuscript organization and revision. C.L. and Y.Z. provided information on the TanSat instrument performance. L.T. provided TanSat in-flight information. M.W. and S.W. provided information on the scientific requirement for data further application. N.L. and D.L. led the TanSat data application. Z.Y. led the TanSat in-flight operation.

**Competing interests**

The authors declare that they have no conflict of interest.

**Acknowledgments**

The TanSat L1B data service was provided by the International Reanalysis Cooperation on Carbon Satellites Data (IRCSD) and the Cooperation on the Analysis of carbon SAtellites data (CASA). The authors thank OCO-2 Team for providing Level-1B data and Level-2 SIF data products. The authors thank the FLUXCOM team for providing global GPP data. The authors thank Google for allowing free use of Google Earth and reproduction of maps for publication. The authors also thank the Landsat/Copernicus team for providing land surface images for Google Earth.

**Financial support**

This work has been supported by the National Key R&D Program of China (Grant No. 2021YFB3901000), the Key Research Program of the Chinese Academy of Sciences (ZDRW-ZS-2019-1), and the Youth Program of the National Natural Science Foundation of China (Grant No. 41905029, Grant No. 42105113).

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
