# Peer review of "Retrieval of Solar-induced Chlorophyll Fluorescence from Satellite 1 Measurements: Comparison of SIF between TanSat and OCO-2 2"

_Atmospheric Measurement Techniques, 2021_

## Author Response (AR1)

Reply RC1

General comment

The manuscript of Yao et al. presents a DOAS-like retrieval algorithm to estimate the amount of solar-induced chlorophyll fluorescence (SIF) from TanSat measurements. For a technical paper, the actual retrieval algorithm is insufficiently well explained and results would not be reproducible. The SIF retrieval strategy including Eq. (2) is well documented in the literature. However, it remains unclear how the only novelty, that is the surface pressure to model residual O2 absorption, is incorporated and affects the retrieval. Perhaps a Figure and sensitivity analysis could illuminate the importance and necessity of this state vector element. There is some confusion as to when the official OCO-2 data set (IMAP-DOAS) and when the IAPCAS data are used. While I think it is a good idea to compare the performance of the proposed retrieval algorithm (Sect. 2.4) based on OCO-2 L1B data, it is inappropriate to use the IAPCAS OCO-2 data in Fig. 3 (where it is not clear which OCO-2 SIF data has been used), 4, and 5 (again not clear which OCO-2 data set has been used). In fact, it may even disguise shortcomings. For example, the difference maps in Fig. 4 suggest that there is considerably less coverage for OCO-2, while I am certain that the official data provides a better coverage. For Figs. 3, 4 & 5, the official OCO-2 data is urgently needed to evaluate the retrieval performance objectively.

Reply: The algorithm part was modified in the revision. The retrieved OCO-2 SIF data and official OCO-2 SIF data were clarified before usage in the revision. We used both the two OCO-2 SIF data in section 2.4 for the algorithm test. The official OCO-2 SIF data was used in the orbit comparison with TanSat. The retrieved OCO-2 data was compared with the TanSat SIF data on the global scale, which was shown in Figure 3. However, the official OCO-2 SIF data was used as a reference for global SIF validation and it was not shown in the manuscript. In the comparison of SIF-GPP, the official OCO-2 SIF data was used for correlation estimation. The global distribution of the SIF difference represents the gridded TanSat and OCO-2 observation data pairs, which are affected by the data volume and distribution of the two satellites, and the actual effective data pairs are less than the OCO-2 data volume, which leads to limited coverage.

As the authors cite Yao et al., 2021 with the same topic and a substantial overlap in co-authors, it is unclear to me what would justify another publication, especially when considering the shortcomings of the present manuscript.

Reply: The previous paper mainly introduced the TanSat SIF product and made a comparison between the two TanSat SIF products by different algorithms. The result shows a regional bias between the two SIF products in different seasons. This article details the IAPCAS/SIF algorithm implementation process, and tests the consistency of the retrieval results with OCO-2 data products by data comparison on different scales, focusing on the collaboration observation of different satellite missions and comprehensive usage of multi-satellite products. The physical-based SIF retrieval method is

Minor comments:
L43: Middle → Medium

Reply: it was modified.

L90: spectrum -> spectral

Reply: it was modified.

L143 What is "side radiance"?

Reply: it was modified in the revision and it means the radiance outside the absorption features in the micro-window.

L149: significant $\rightarrow$ systematic

Reply: it was modified.

L184 Point 2: I believe the opposite is meant here, continuum level radiances outside the range of 15-200 W/m²/µm/sr

Reply: it was modified in the revision.

L196 & Table 1: The authors mention that there is a small bias between the official OCO-2 product and the results of their own retrieval algorithm. I suggest to add the intercept and slope to Table 1, so that the reader can come to their own judgement as to how well the retrieval algorithm performs.

Reply: The intercept and slope were added to Table 1.

L256-259: The described quantity should not be referred to as "retrieval uncertainty". In fact, this is the standard error of the mean and is a measure of retrieval error plus natural variability.

Reply: it was modified in the revision.

Reply RC2

The manuscript by Yao et al. presents the retrieval of SIF from TanSat satellite measurements and compares the retrieved TanSat SIF to OCO-2 SIF and GPP data. This study covers a great mission and dataset and the topic is important for the scientific community. The manuscript is written in a concise way, however, there are some open questions which are not/partly addressed. I recommend it to be accepted after the following issues are addressed.

General comments:

- The algorithm presented in this study has already been partly shown, tested and optimized in a previous study. In the present manuscript, the authors describe the used algorithm, but do not explain what is new/different compared to other existing SIF retrieval algorithms. They directly compare the SIF results to other SIF measurements. As this is a technical journal, I think it would be important to have more insight on the used SIF retrieval algorithm, particularly how it compares to existing algorithms, where the differences are, why a new algorithm is used etc. To what extend have the points mentioned for example in Parazoo et al., 2019 (https://doi.org/10.1029/2019JG005289) been considered when comparing different SIF satellite products? Why did the authors choose OCO-2 SIF and not for example TROPOMI SIF as a comparison?

Reply: The IAPCAS/SIF algorithm introduced in the paper is based on the simplified physical model. The main optimization is the usage of a scale factor to correct the influence of O2 column absorption induced by the the uncertainty of surface pressure in the inversion state vector to reduce the interference of the O2 absorption line on the SIF signal. It was clarified

in the revised version. This algorithm is developed for the TanSat data produce and application. The currently commonly used SIF inversion algorithms include the data-driven algorithm and the DOAS algorithm, but these two algorithms are not accessible, and the SVD-driven algorithm is greatly affected by training samples. Therefore, the main purpose of the paper is to establish a reliable algorithm to obtain SIF data from TanSat satellites to provide SIF products. Based on the TanSat SIF data products obtained by the data-driven algorithms, it is found that there is a seasonal deviation between the TanSat SIF products obtained by the SVD data-driven algorithm and the TanSat SIF products based on the physical model, so the OCO-2 product is used for further verification of the algorithm. The reason for adopting OCO-2 is that OCO-2 and TanSat have similar observation modes, including scanning method, transit time, spatial resolution, spectral resolution, spectral range. The similarities mean that the SIF product from the two missions can be directly compared. However, TROPOMI and TanSat have a large difference in spatial resolution and spectral resolution, and the SIF retrieval method and the spectral fitting range are also different. To perform direct algorithm verification and product consistency analysis, the usage of OCO-2 data as a reference is the most direct and effective way.

- The order of the introduction and the transitions from one paragraph to the next are sometimes hard to follow. The TanSat satellite is mentioned in a different paragraph than the other SIF satellites but without highlighting the differences. It is also not mentioned that first TanSat SIF maps already exist (Du et al., 2018) and why a new algorithm has to be used. The scientific/ research questions are missing in the introduction.

Reply: The order of the introduction is modified in the revision. The SIF product by SVD method was explained and the reason for the development of the IAPCAS/SIF algorithm and the research question was also clarified in the revision.

- Besides global maps, the authors present results from a sample region using maps and a correlation plot. However, besides this visual comparison, I think a SIF timeseries of the chosen dataset in this sample region in comparison to OCO-2 SIF is very helpful and should be discussed.

Reply: the SIF time series is a significant way to evaluate the stability of the long-term data consistency, but due to the difference in satellite observation time and location, as well as the difference in the land cover types, it is hard to form matching observation pairs for effective time series comparison.

Minor/technical comments:

L1: I would add something like satellite/spaceborne/ from space etc. to the title

Reply: the title was modified to 'Retrieval of Solar-induced Chlorophyll Fluorescence from Satellite Measurements: Comparison of SIF between TanSat and OCO-2'.

L18: What is a sensitive instrument? Please clarify shortly.

Reply: it was clarified in the revision by 'sensitive instruments with high SNR and spectral resolution'.

L21: Globally or for which location and resolution?

Reply: It was clarified in the revision. The data over the global were processed at the sounding scale.

L24: gridded

Reply: It was modified.

L25: Specify what the official OCO-2 SIF product is and what the difference is to L22/ the product retrieved in this study

Reply: The official OCO-2 SIF product used in the paper is the OCO2_Level 2_Lite_SIF.8r, and it contains SIF for each sounding daily. The data is provided by the OCO-2 Science team and used to test the retrieval algorithm by comparing it with the retrieved OCO-2 SIF results. The seasonal difference between the two OCO-2 SIF product is less than 0.2 0.2 W m$^{-2}$ µm$^{-1}$ sr$^{-1}$ in a 1° × 1° grid.

L25: seasonally-gridded

Reply: It was modified.

L27: Where does this GPP data come from (Ground-based/globally, spatio-temporal resolution etc.). What is the result of the comparison?

Reply: The FLUXCOM gross primary productivity (GPP) used here is the monthly global gridded flux products in a 0.5° × 0.5° grid, which are calculated from ground-based FLUXNET measurements and mean seasonal cycles according to MODIS data and daily meteorological information with a machine learning method. The relationship between annual averaged SIF products and FLUXCOM gross primary productivity (GPP) for six vegetation types shows that the SIF data from the two satellites have the same potential in quantitatively characterizing ecosystem productivity.

L36: remove ;

Reply: It was removed.

L46: ··· and Frankenberg et al. 2011

Reply: It was added.

L94: what about OCO-2 data?

Reply: it was modified in the revision. The SVD method was applied to several satellite missions, including GOSAT, OCO-2, TanSat, and S5p/Tropomi.

L98: What is the major outcome of this previous study?

Reply: The previous study introduced the TanSat SIF product by using the physical model method and compared it with the SVD method based TanSat SIF data. The comparison shows that the two SIF products are relatively consistent on the seasonal scale, but there are obvious regional deviations. Due to the different biases in four seasons, the regional biases could be caused by different training samples in the SVD method.

L100: What are the research questions for this study?

Reply: To verify the reliability of the IAPCAS/SIF algorithm and further test the potential of different satellites in a comprehensive analysis of SIF, this study detailed the IAPCAS/SIF algorithm and made a SIF comparison between TanSat and OCO-2 at both sounding and global scales.

L101: a bit out of context, what other products are availabe and why this selection? Maybe move this selection to the retrieval methd; Space between number and unit.

Reply: We use the SIF signal at 757 nm because SIF emission intensity in the 757nm micro-window is stronger due to being closer to the SIF emission peak, and the interference from other absorption lines is weaker than that in the 771 nm micro-window. The 757 nm SIF is more stable. The selection is moved to the method part.

L106: Why is the wavelength window name (757 nm, 771 nm) not part of the shown

wavelength range (758.3-759.2 nm, 769.6 – 770.4 nm)?

Reply: Following the traditional rules, we keep the notation of the two micro-windows as the 757 nm window and the 771 nm window respectively for consistency.

L116: reference missing

Reply: it was added in the revision.

L146: Not all readers are XCO2 retrieval experts, please explain the complexity and why this approach was selected.

Reply: it was modified in the revision.

L171: Specify the footprints in both panels.

Reply: It was modified in the revision.

L188: first

Reply: It was modified.

L191: add reference for this retrieval approach; for which spatial and temporal resolution and location?

Reply: The reference was added and the lite product was introduced briefly in the revision. The lite file provides SIF measurement of each sounding daily over the globe, and hence the data spatial-temporal resolution is the same as the sounding pixel.

L196: applied to

Reply: It was modified.

L197: remove ‚remained‘

Reply: It was removed.

L207: check wording

Reply: It was modified.

L221: From which satellite are the SIF measurements in (a)? ‚evergreen‘ instead of ‚evergrenn‘

Reply: Figure 3(a) shows the SIF from both OCO-2 and TanSat for a whole view of the satellite measurements. The legend in figure 3 was modified.

L224: check wording

Reply: it was modified in the revision.

L238: The TanSat SIF data shown here is from 2017-2018; ground-based SIF measurements from different stations globally are already available for this time.

Reply: This paper focuses on the comparison between space-based SIF products. The researches on ground SIF and satellite SIF measurements will be held in future researches.

L248: Which instrument performances are meant here? Please explain.

Reply: The instrument performance difference contains SNR and instrument respond functions, which is represented by the different structural characteristics of the bias curves.

L298-306: Are there and what are the differences between the TanSat-GPP and OCO-2-GPP correlation? This is shown in the Figure, but not mentioned in this discussion part.

Reply: it was explained in the revision. For shrubland and grassland, the slope of OCO-2 SIF with GPP is higher than that of TanSat and has a worse correlation. For forests, OCO-2 SIF present a better correlation with GPP, especially in the needle leaf forest.

L315: What are the major improvements/ changes from TanSat to TanSat-2?

Reply: TanSat-2 intends to build a constellation of six satellites for atmospheric concentration observation with a high spatial-temporal resolution to support the carbon neutrality goal and researches on global change. The target gases of TanSat-2 will cover more kinds of gases,

including $CO_2$, $CH_4$, CO, and NOx. TanSat-2 will also provide SIF measurement. The improvement of TanSat-2 was partly supplemented in the revision.

---

## Author Response (AR2)

Comment and Reply

Some points have been clarified by the authors in the revised manuscript, but some points still remain unclear. These points still need to be revised:

General points:
- The intro is still confusing: What is the main difference and new part of the IAPCAS/SIF algorithm compared to the mentioned existing algorithms? Why is it compared to OCO-2 in this study and to what extend? This was already summarized and addressed in the authors' reply. I think it would be good to add these explanations to the manuscript too. The authors should also clearly highlight that the new algorithm has already been used and tested in Yao et al 2021 and should also explain why the algorithm is now presented in this manuscript and/or what is different to Yao et al 2021.
Reply: The information was added and clarified in the revision.

- L141-198: What exactly is the new part of the IAPCAS/SIF algorithm and how is this incorporated and tested? Are there any sensitivity studies? As this is a technical journal, I think it is important to have more insight on the used algorithm.
Reply: The IAPCAS/SIF algorithm introduced in the paper is based on the simplified physical model. The main optimization is the usage of a scale factor to correct the influence of O2 column absorption induced by the uncertainty of surface pressure in the inversion state vector to reduce the interference of the O2 absorption line on the SIF signal. Sensitivity studies were not performed, but our first version of SIF product retrieved without scale factor in state vector shows a worse global distribution and magnitude. The content was modified in the revision: "In other physical-based retrievals, the surface pressure data of the European Centre for Medium-Range Weather Forecasts (ECMWF) is usually used as the true surface pressure to simulate the molecular absorption cross-section. However, there is still a difference between the true surface pressure and the model surface pressure, so we introduced a factor here to reduce the influent of the inaccurate surface pressure."

- L400-402: There is little discussion of the SIF-GPP results. What are potential explanations for these differences and findings? What do we learn from this? The SIF-GPP results are also not mentioned in the conclusions at all.
Reply: For the same vegetation type, the SIF-GPP correlations of the two satellites are relatively similar, indicating that the two satellite SIF products have similar GPP characterization capabilities. It shows that the comprehensive application of different satellite products has strong feasibility. For different vegetation types, the SIF-GPP correlations for both satellites were significantly different, indicating that there were differences in the ability of SIF to characterize different vegetation GPP. A detailed explanation for SIF-GPP correlations was added in the revision.

Minor points:

L18: ‚signal-to-noise ratio' instead of ‚SNR' in the abstract

Reply: it was modified in the revision.

L21: What does ‚at sounding scale' mean?

Reply: the SIF signal was retrieved for each sounding and it was modified in the revision.

L22-24: It is still not entirely clear for the reader which SIF algorithm (‚the SIF retrieval algorithm') is used here. Could you give your new SIF algorithm a name (IAPCAS/SIF?)?

Reply: "the physical-based algorithm" here was replaced by "the IAPCAS/SIF algorithm" in the revision.

L28: annually

Reply: it was modified in the revision.

L375: Is the GPP resolution 1°x1° or 0.5°x0.5° as stated in the author reply?

Reply: the spatial resolution of GPP product is 0.5°x0.5° and it was resampled to 1°x1° grid-cell for comparison with SIF in the study. It was clarified in the revision.

---

## Author Response (AR3)

**Reply on the Associate Editor**

the manuscript still needs some clarificiations and some polishing in terms of English language issues. I provide an annotated manuscript suggesting various edits. Please consider also consulting an English native speaker.

Reply: Thanks for your kind annotation. The manuscript was revised according to the comments, and some modifications were made in Line 247-257, 311-314, 353-356, 411.